# Effects of Varying Levels of Wheat Bran Dietary Fiber on Growth Performance, Fiber Digestibility and Gut Microbiota in Erhualian and Large White Pigs

**DOI:** 10.3390/microorganisms11102474

**Published:** 2023-10-01

**Authors:** Taoran Du, Pinghua Li, Qing Niu, Guang Pu, Binbin Wang, Gensheng Liu, Pinghui Li, Peipei Niu, Zongping Zhang, Chengwu Wu, Liming Hou, Mette Skou Hedemann, Qingbo Zhao, Ruihua Huang

**Affiliations:** 1Key Laboratory of Evaluation and Utilization of Livestock and Poultry Resources (Pig) of Ministry of Agriculture and Rural Affairs, Institute of Swine Science, College of Animal Science & Technology, Nanjing Agricultural University, Nanjing 210095, China; 2017205008@njau.edu.cn (T.D.); lipinghua718@njau.edu.cn (P.L.);; 2Huaian Academy, Nanjing Agricultural University, Huaian 223005, China; 3Institute of Animal Husbandry and Veterinary Science, Shanghai Academy of Agricultural Sciences, Shanghai 201403, China; 4Depatment of Animal Science, Aarhus University, 8830 Tjele, Denmark

**Keywords:** dietary fiber, Erhualian pigs, fiber tolerance, gut microbiota, growth performance

## Abstract

To evaluate the tolerance of a high-fiber diet in Erhualian pigs (Er-HL), the present investigation systematically investigated the ramifications of varying wheat bran fiber levels, specified as total dietary fiber (TDF) values of 14.07%, 16.32%, 17.99%, and 18.85%, on growth performance, fiber digestibility and gut microbiota in Er-HL, large Large White pigs (L-LW, the same physiological stage as the Er-HL) and small Large White pigs (S-LW, the same body weight as the Er-HL). Our results revealed that fiber levels exerted no discernable impact on growth performance (average daily feed intake (ADFI), and average daily gain (ADG)) of Er-HL (*p* > 0.05). Conversely, L-LW exhibited a decrease in ADFI and ADG with increasing fiber levels (*p* < 0.05). Notably, the apparent total tract digestibility (ATTD) of various fiber components, including neutral detergent fiber (NDF), acid detergent fiber (ADF), hemicellulose, TDF and insoluble dietary fiber (IDF), in Er-HL were significantly higher than those in S-LW and L-LW irrespective of diets (*p* < 0.05). The ATTD of cellulose and hemicellulose in Er-HL significantly decreased with increasing fiber levels (*p* < 0.05), yet remained statistically indifferent when comparing the 7%-wheat-bran-replaced diet (7% WRB, TDF 16.32%) to the basal diet (TDF 14.07%) (*p* > 0.05). The cecal microbiota of Er-HL had higher richness estimators (Chao1 and ACE) than those of S-LW and L-LW irrespective of diets (*p* < 0.01). Breed serves as a pivotal determinant in shaping swine gut microbiota. Thirteen genera were selected as the key bacteria related to high fiber digestibility of Er-HL. Further functional examination of these key genera elucidated an enrichment of pathways pertinent to carbohydrate metabolism in Er-HL samples compared with S-LW and L-LW samples. In summary, Er-HL exhibited high-fiber tolerance both in terms of growth performance and fiber digestibility compared with Large White pigs. Specifically, the ATTD of NDF, ADF, hemicellulose, IDF and TDF were significantly higher in Er-HL compared with L-LW and S-LW, irrespective of diets. Fiber level exerted no discernable impact on growth performance (ADFI, ADG) and the ATTD of fiber (NDF, ADF, IDF and TDF) in Er-HL. The optimum fiber level of the Er-HL was identified as 7% WRB (TDF 16.32%). Thirteen genera were ascertained to significantly contribute to high fiber digestibility of Er-HL, correlating with an enhancement of carbohydrate metabolism pathways.

## 1. Introduction

At present, commercial pig breeds such as Large White and Landrace dominate the Chinese pig market [1]. These breeds are characterized by accelerated growth rates and elevated lean meat yield, and primarily subsist on a diet of corn and soybean meal. However, an escalating competition for these feed grains between human populations and commercially raised pigs is being observed, particularly in nations constrained by limited availability of corn and soybean meal, such as China. Contrarily, alternative feed sources such as wheat bran, rice bran, and other non-conventional feed resources exhibit not only a more economical price point but also a reduced carbon footprint. Numerous studies have scrutinized the potential advantages of replacing imported feed components, particularly those of soybean origin that accrue significant environmental costs from transport and land use alteration, with locally sourced ingredients [2]. Optimization of agricultural practices and judicious formulation of swine diets could mitigate the overall carbon footprint of these production systems [3].

The implementation of these non-conventional feed resources in pig diets is predominantly restricted due to their high dietary fiber composition [4]. Nonetheless, local pig breeds demonstrate a heightened tolerance and exhibit superior digestive capabilities for dietary fiber compared with commercial pig breeds [5,6]. Consequently, burgeoning interest has been manifested in the evaluation of non-conventional feed resources inputs in diets tailored for local pig breeds and novel pig hybrids possessing local genetic lineage, aimed at curtailing feed-related expenditures.

It is well-established that neither the porcine stomach nor the small intestine synthesizes enzymes capable of hydrolyzing dietary fiber. The degradation of dietary fiber primarily occurs via microbial hydrolysis in the large intestine [7]. Specifically, cellulolytic bacteria produce cellulases, which act on fiber substrates to yield monosaccharides and oligosaccharides. These smaller carbohydrates are subsequently fermented by the gut microbiota into short-chain fatty acids (SCFAs) such as acetic acid, propionic acid and butyric acid. These SCFAs can be absorbed across intestinal epithelial barriers and distributed via systemic circulation to various tissues. Notably, colonic SCFA production contributes to 5–28% of the energetic requirements for growing pigs [8].

Previous research endeavors have primarily centered upon fiber-degrading bacteria within the rumen of ruminants, thereby laying foundational knowledge applicable to monogastric organisms [9]. The categorization of fiber-degrading bacteria is often based on their substrate specificity for cellulose, hemicellulose or pectin [7,10]. Although earlier methodologies relied upon culture separation or in vitro fermentation techniques to identify porcine intestinal fiber-degrading bacteria [11,12], these approaches are fraught with limitations. Notably, over 99% of bacteria cannot be purified under laboratory conditions, and in vitro fermentation systems inadequately emulate the in vivo intestinal environment [13]. Additionally, extant literature indicates differential microbial diversity and abundance across distinct intestinal segments. Moreover, there exists a defined sequence by which intestinal microorganisms partake in fiber degradation, thereby rendering artificial simulations inherently imprecise [14,15]. Recent advances in 16S rRNA gene sequencing offer innovative avenues for systematic offer innovative avenues diversity and richness within the intestinal microbiota. Certain studies have elucidated that the observed variations in microbial composition among the different breeds may be attributable to the breed characteristics [16,17].

Moreover, research conducted on the gut microbiota across various development stages of swine has revealed that the apparent digestibility of crude fiber in Sutai pigs positively correlates with age, and specific bacterial genera have been implicated in this phenomenon [18]. Numerous variables, including dietary fiber levels, breed, and fiber source, modulate both the composition of fiber-degrading bacteria and the overall process of fiber digestion and absorption in pig [7]. While several studies have evidenced higher fiber digestibility in indigenous pig breeds relative to commercial lean pig breeds [19,20,21], specific bacterial communities driving this high fiber digestibility remain unidentified. Furthermore, conflicting reports suggest comparable fiber digestibility between local pig breeds and commercial pig breeds, potentially attributable to variations in body weight or physiological stage across these studies [22,23]. 

Erhualian pigs are distributed around the Taihu Lake region in the lower Yangtze River valley of China. Erhualian pig is a typical representative of local Chinese pig breeds. Renowned for their high prolificacy [24], they additionally display traits of fiber tolerance and excellent meat quality [25]. Nonetheless, advancements in the research and development of superior genetic traits within Chinese local breeds have proceeded at a relatively attenuated pace.

The objective of the present study is to assess the fiber tolerance and identify the fiber-degrading bacterial taxa associated with high fiber digestibility of the Chinese Erhualian pig breed. Additionally, the study aims to delineate differences in growth performance, apparent fiber digestibility, and gut microbiota structure between Erhualian and Large White pigs, the latter being chosen for uniformity in body weight and physiological stage.

## 2. Materials and Methods

### 2.1. Animals and Experimental Diets

To investigate the tolerance to elevated dietary fiber content in Erhualian pigs, a parallel comparison was executed against Large White pigs, sharing analogous physiological stages and body weight. The methodological construct of the current investigation is expounded in a previously published work by our team [26]. Considering the Erhualian pigs reached puberty early, as evidenced by prior studies [25,27], the average body weight of sows in their third parity served as a surrogate metric for mature body weight. The average mature body weight of Erhualian pigs is 150 kg, whereas that of Large White pigs is 240 kg [28]. When the body weights of these two pig breeds reach the same percentage of their mature body weights, they are considered to be at the same physiological stage [26]. Twenty-four Erhualian fattening barrows (Er-HL, 26.7% of mature body weight, approx. 40 kg), 24 large Large White fattening barrows (L-LW, the same physiological stage with the Er-HL, 26.7% of mature body weight, approx. 65 kg) and 24 small Large White fattening barrows (S-LW, the same body weight as the Er-HL, approx. 40 kg) were distributed across four distinct dietary interventions in a completely randomized block design. The experimental configuration comprised 12 groups, each incorporating four treatment regimens for Er-HL, L-LW, and S-LW, with six replicates per group [26]. A basal diet, along with three subsequent experimental diets, was employed. The basal diet, characterized by neutral detergent fiber (NDF) at 12.84%, total dietary fiber (TDF) at 14.07%, and insoluble dietary fiber (IDF) at 13.65% was devised in accordance with the Chinese “Feed Standard of Swin (NY/T 65-2004)” [29]. Amounts of 7%, 14%, and 21% wheat bran were used to equivalently replace the basal diet with the experimental diets (refer to Appendix A). The TDF concentrations of the basal diet (Basal), 7%-wheat-bran-replaced basal diet (7% WRB), 14%-wheat-bran-replaced basal diet (14% WRB), and 21%-wheat-bran-replaced basal diet (21% WRB) were recorded as 14.07%, 16.32%, 17.99%, and 18.85%, respectively (refer to Appendix A). The 6 pigs in each group were housed in a pen measuring 2.5 m in width and 5.3 m in length. Osborne Testing Stations System (OTSS, Osborne Industries, Inc., Osborne, KS, USA) were utilized for recording of average daily feed intake (ADFI) and weight gain. The pigs were allowed ad libitum access to water and pelleted feed throughout the entirety of the study duration. During the experimental course, climate control measures, including wet curtain cooling fans and air-conditioning systems, were employed to maintain ambient humidity and temperature at approximately 65% relative humidity (RH) and 25 °C. Daily inspections were conducted to ensure constant environmental parameters.

For a pre-experimental acclimatization period of 10 days (d −10 to 0), subjects were administered the basal diet. Subsequent to this period, a 28-day experimental period (d 0 to 28) was initiated, during which varying levels of dietary fiber were introduced via the experimental diets. Throughout the study duration, it is pertinent to note that all subjects remained in optimal health, thereby obviating the necessity for antibiotic interventions. All experimental procedures and the handling of animals were carried out according to the Guidelines the Care and Use of Laboratory Animals prepared by the Nanjing Agricultural University Animal Welfare and Ethics Committee (Certification No. SYXK (Su) 2022-0031).

### 2.2. Animals Slaughter and Samples Collecting

Diet samples were collected at the onset of the study and cryogenically stored at −20 °C. On d 28, fresh fecal samples were procured from within the barn premises at the end of the experiment period for determination of apparent digestibility. Each 200 g aliquot of fecal sample was mixed with 15 mL of 10% sulfuric acid solution, ensconced in polyethylene bags, and cryogenically stored at −20 °C pending the evaluation of nutritional apparent digestibility. Subsequent to these procedures, all porcine subjects underwent weighing prior to their transit to a certified abattoir at 08:00. The euthanasia procedures were bifurcated into two discrete temporal batches: the first transpiring at 13:00., and the second at 18:00. Following electrical stunning, exsanguination, depilation, and evisceration were conducted in stipulated standard operating protocols. Cecal content samples were promptly harvested, instantaneously subjected to snap-freezing via liquid nitrogen, and stored at −80 °C in anticipation of DNA isolation and 16S rRNA gene sequencing.

### 2.3. Experimental Chemical Analysis

Both diet and fecal samples underwent a drying process at a constant temperature of 65 °C until reaching a state of stable mass. Subsequent grinding through a 0.45 mm sieve facilitated subsequent analyses. Acid-insoluble ash (AIA) served as an indigestible marker in the evaluation of apparent total tract digestibility (ATTD) of the dietary components, as delineated by AOAC guidelines (AOAC 942.05). The NDF and the acid detergent fiber (ADF) content carried out using the ANKOM A200 Fiber Analyzer (ANKOM Technology, Macedon, NY, USA) filter bag technique based on AOAC 962.09, while the content of cellulose and hemicellulose were calculated based on NDF and ADF. The content of IDF, soluble dietary fiber (SDF) and TDF were measured with the Megazyme total dietary fiber assay procedure (catalogue no. K-TDFR 05/12, Wicklow, Ireland). The ether extract (EE) content was measured using a Soxhlet equipment (Xinsande instrument Co., Ltd., Hebi, China) based on the Soxhlet extraction method, AOAC 920.85. The crude protein (CP) content was measured using a Kjeltec 8400 analyzer unit (Foss, Hoganas, Sweden) based on the Kjeldahl method, AOAC 984.13. SCFA concentrations in the cecum content samples were determined using GC-14B gas chromatography (Shimadzu, Kyoto, Japan) with a flame ionization detector and a Capillary Column (Agilent DB-1701, 30 m × 0.32 mm × 0.25 μm film thickness, Agilent Technologies, Waldbronn, Germany). The sample preparation for SCFA analysis was congruent with methods elucidated by Zhang [30]. The injector and detector temperature was set at 180 °C/180 °C, the column temperature was heated at a rate of 20 °C/min from 60 °C to 220 °C and maintained for 1 min, and the gas flow rate was 30 mL/min. Analyte peaks were identified by comparing their retention times with standards of acetate, propionate, butyrate, isobutyrate, valerate, isovalerate and metaphosphoric–crotonic acid. Standard curves were constructed and the quantification of the SCFAs was made based on individual calibration curves.

### 2.4. 16S rRNA Gene Sequence and Bioinformatics Analysis

The selection of samples for the 16S rRNA gene sequence was contingent upon the growth performance and apparent digestibility of fiber. Groups administered Basal diet, 7% WRB and 21% WRB across Er-HL, L-LW and S-LW categories were earmarked for the scrutiny of fiber-level impacts on gut microbial diversity and community structure. The E.Z.N.A.^®^ soil DNA Kit (Omega Bio-tek, Norcross, GA, USA) was used to extract the total genomic DNA of micriobiota from the cecum content samples. The quantity and quality of DNA were measured using a NanoDrop 2000 spectrophotometer (NanoDrop 2000, Thermo, Waltham, MA, USA) to make sure the absorption ratio (260/280 nm) was within 1.8–2.0, which means the DNA was deemed to be of sufficient purity to be used for subsequent analyses. For Illumina-based sequencing, the V3–V4 hyper-variable region of the 16S rRNA gene was amplified by PCR. The primer 338F (5′-ACTCCTACGGGAGGCAGCAG-3′) and 806R (5′-GGACTACHVGGGTWTCTAAT-3′) targeting the bacteria 16S V3-V4 regions were selected for the microbiota analysis [31]. Agarose gel electrophoresis ascertained the singularity and specificity of the product. Tripartite amplification products derived from identical samples were amalgamated, and subsequently subjected to purification via Agencourt AMpure XP nucleic acid purification magnetic beads in isovolumetric proportions. The purified products were used for sequencing by synthesis. Barcoded V3-V4 amplicons were sequenced utilizing the Illumina MiSeq platform (Illimina, San Diego, CA, USA) at Shanghai Majorbio Bio-pharm Technology Co., Ltd. (Shanghai, China) according to the manufacturer’s instructions.

Raw sequence reads were truncated by excising the barcode and primer regions, and subsequently merged utilizing FLASH software [32]. These reads were then subjected to trimming, filtration, alignment, and taxonomic classification via the Mothur software suite [33]. Operational taxonomic units (OTUs) were categorized at a similarity threshold of ≥97% by employing the UPARSE pipeline [34]. Taxonomic assignments were ascertained via the Ribosomal Database Project (RDP) classifier employing a confidence cutoff value of 0.8 [35,36]. 

### 2.5. Calculations and Data Analysis

The content of hemicellulose and cellulose were calculated as follows:(1)$$\mathrm{Hemicellulose}=\mathrm{NDF}-\mathrm{ADF}\;\mathrm{Cellulose}=\mathrm{ADF}-\left(\mathrm{Ash}+\mathrm{Lignin}\right)$$

The ATTD of each sample was calculated by the following equation:(2)$${\mathrm{CAD}}_{D}\left(\%\right)=100\times \frac{1-\left({\mathrm{DC}}_{F}\times {\mathrm{AIA}}_{D}\right)}{{\mathrm{DC}}_{D}\times {\mathrm{AIA}}_{F}}$$
where CAD_D_ is the coefficient of the apparent digestibility of dietary components in the assay diet; DC_F_ is the dietary component concentration in feces; AIA_D_ is the AIA concentration in the assay diet; DC_D_ is the dietary component concentration in the assay diet; and AIA_F_ is the AIA concentration in feces [18].

Data pertaining to growth performance were gleaned from the OTSS. Subsequent analyses of growth performance, fiber digestibility, and alpha diversity indices within bacterial communities were orchestrated in accordance with a factorial design. This involved a stratification of four dietary fiber levels of fiber diets (Basal diet, 7% WRB; 14% WRB, and 21% WRB) and three distinct pig populations (Er-HL, L-LW and S-LW). The analyses were conducted employing the PROC MIXED procedure within the SAS 9.0 statistical software package [37].
Y_ijkl_ = μ + D_i_ + B_j_ + A_k_ +W_l_ + B_j_ × D_i_ + e_ijkl_(3)
where μ is the overall mean, D_i_ is the fixed effect of diets (i = 1–4), B_j_ is the fixed effect of breeds (j = 1–3), A_k_ is the fixed effect of slaughter batches (k = 1–2), W_l_ is the covariate effect of final pig body weight (l = 1–72), B_j_ × D_i_ is the interaction of breads on diets, and e_ijkl_ is the residual error.

Significance was declared when *p* < 0.05, and tendency was discussed when *p* < 0.10. Moreover, breed × diet interaction effects were scrutinized through the PROC MIXED procedure of SAS 9.0, where discernable. Comparative analyses of fiber digestibility across the 4 diets, within homogenous breed categories, were performed via One-way Analysis of Variance (ANOVA) employing Statistical Product and Service Solutions (SPSS) 20.0 software (International Business Machines Corp., Armonk, NY, USA) [38]. 

Factors influencing microbiome composition were assessed via Permutational Multivariate Analysis of Variance (PERMANOVA) [39]. The Principal Coordinates Analysis (PCoA) of beta-diversity within microbial communities, predicated upon unweighted UniFrac distance metrics, was executed employing Bray–Curtis dissimilarity measures for all sample comparisons [40]. Microbiota variances were evaluated using the Linear Discriminant Analysis Effect Size (LEfSe) methodology [41]. Furthermore, potential functional attributes inherent to bacterial communities were probed utilizing the Phylogenetic Investigation of Communities by Reconstruction of Unobserved States 2 (PICRUSt2) algorithm [42]. Differences with *p* < 0.05 were considered as statistically significant, and a tendency was considered at 0.05 ≤ *p* < 0.10. 

## 3. Results

### 3.1. Effect of Dietary Fiber Level in the Diet and Breed on the Growth Performance

The growth performance of Er-HL, L-LW, and S-LW under the influence of a basal diet and additional experimental regimens incorporating varying levels of dietary fiber are shown in Table 1. A statistically significant breed × diet interaction was manifested in the average daily gain (ADG) and average daily feed intake (ADFI) variables. Irrespective of diets, the initial body weight of Er-HL was significantly lower (*p* < 0.01) compared with the L-LW. Likewise, final body weight and ADG were significantly lower (*p* < 0.01) in Er-HL than in L-LW and S-LW after a 28-day evaluation period. The feed/gain ratio (F/G) values were significantly higher (*p* < 0.01) in Er-HL compared with L-LW and S-LW. Increasing the bran fiber level from Basal to 14% WRB decreased ADFI irrespective of breeds (*p* < 0.05). Compared with the Basal diet group, the 7% WRB and 14% WRB groups exhibited a reduction (*p* < 0.05) in ADG. 

Figure 1 delineates that the ADG and ADFI of L-LW fed with 7%, 14% and 21% WRB decreased significantly as opposed to the basal diet (*p* < 0.05). In contrast, the ADG and ADFI of Er-HL and S-LW were not influenced by the bran fiber level under identical dietary conditions. Furthermore, the F/G of Er-HL fed with 14% WRB was notably reduced in comparison to the 7% WRB group (*p* < 0.05).

### 3.2. Effect of Dietary Fiber Level in the Diet and Breed on the Total Tract Apparent Digestibility

Table 2 displays the ATTD of primary dietary components in Er-HL, L-LW, and S-LW under basal and experimental diets containing differing concentrations of bran fiber. A breed × diet interaction was substantiated with respect to the ATTD of CP. Irrespective of diets, the ATTD of NDF, ADF, hemicellulose, IDF and TDF were significantly higher (*p* < 0.05) in Er-HL compared with L-LW and S-LW. Moreover, the ATTD of CP was significantly higher (*p* < 0.05) in Er-HL and L-LW as compared with S-LW. Increasing the fiber level from Basal to 21% WRB led to an increase in ATTD of SDF of pigs (*p* < 0.05), yet induced a decrease in ATTD of NDF, ADF, cellulose, hemicellulose, IDF, EE and CP of pigs (*p* < 0.05) irrespective of breeds.

The ATTD of fiber in Er-HL, S-LW and L-LW at four fiber levels concentrations is illustrated in Figure 2. The ATTD of NDF (Figure 2A), ADF (Figure 2B), IDF (Figure 2E), TDF (Figure 2G) in Er-HL remained invariant upon elevation of dietary fiber content. Conversely, the ATTD of NDF, ADF, cellulose (Figure 2C), hemicellulose (Figure 2D), IDF, TDF in L-LW decreased from Basal to 21% WRB (*p* < 0.05). Furthermore, the ATTD of cellulose and hemicellulose in Er-HL manifested a significant diminution in 14% WRB and 21% WRB groups compared with the 7% WRB group (*p* < 0.05). However, there were no differences in the ATTD of cellulose and hemicellulose in Er-HL between 7% WRB and basal diet groups. The ATTD of SDF (Figure 2F) escalated concomitantly with an increase in fiber level (*p* < 0.05). Specifically, the ATTD of SDF in Er-HL was higher in the 7% WRB group compared with the basal diet group (*p* < 0.05). Nonetheless, no such disparity in the ATTD of SDF was discerned between the 7% WRB and basal diet groups in L-LW and S-LW.

### 3.3. Effect of Dietary Fiber Level in the Diet and Breed on Gut Microbial Diversity and Community Structure

Section 3.3 delineates the ramifications of dietary fiber level and breed on gut microbial diversity and community structure as assessed via 16S rRNA gene sequencing. Predicated upon the ATTD of fiber, cecal content samples from the Basal, 7% WRB, and 21% WRB groups of Er-HL, L-LW, and S-LW were scrutinized. A total of 6,708,767 raw sequences was procured from a pool of 54 samples. After quality control, 3,218,841 high-quality sequences were obtained. The average sequence length was 440 bp. The Coverage index was 99.6%.

Table 3 enumerates the alpha diversity indexes. The Simpson diversity within cecal microbiota showed breed × diet interaction. Irrespective of diets, the cecal microbiota of Er-HL pigs had higher (*p* < 0.01) richness estimators (Chao1 and ACE) relative to their L-LW and S-LW counterparts. Moreover, the cecal microbiota of Er-HL and L-LW had higher (*p* < 0.01) species diversity (Shannon) in comparison to S-LW. Concomitant with an increment in fiber level from Basal to 21% WRB, a decrease in cecal microbial richness estimators (Chao1 and ACE) was discernible (*p* < 0.05), irrespective of breeds.

Figure 3A portrays the outcomes of multivariate grouping Principal Component Analysis (PCA) on cecal microbiota, indicating that the microbial profiles of Er-HL were distinctly clustered apart from those pertaining to L-LW and S-LW; diet-induced groupings were similarly segregated. Further, the PCoA illustrated in Figure 3B reveals that the community structure of Er-HL cecal samples were significantly divergent from those of L-LW and S-LW, substantiated by an Analysis of Similarities (ANOSIM) *p*-value of 0.001.

Table 4 elucidates the determinants influencing the swine cecal microbiota as uncovered through Permutational Multivariate Analysis of Variance (PERMANOVA). Examination of variables including diet, breed, body weight (BW), and average daily feed intake (ADFI) indicated breed as the paramount factor sculpting the swine cecal microbiota (*p* < 0.05), whereas diet, BW and ADFI did not exert a significant impact.

### 3.4. Effect of Dietary Fiber Level in the Diet and Breed on Phyla and Genera of Gut Microbiota

This investigation explores the influence of dietary fiber level and breed on the taxonomic delineation, specifically at the phyla and genera strata, of the porcine cecal microbiota. At the taxonomic rank of phylum, sequence data corresponding to cecal microbiota were allocated to 18 phyla. Predominantly, *Firmicutes* and *Bacteroidetes* constituted in excess of 85% the cumulative sequence data (Figure 4A). A total of 312 genera were taxonomically classified, among which *Lactobacillus* and *Prevotella_9* were markedly dominant, accounting for relative abundances of 22% and 13%, respectively (Figure 4B).

Figure 5A–F depict variations in cecal microbiota across distinct breeds as determined by LEfSe. At an alpha level of *p* < 0.05 and a Linear Discriminant Analysis (LDA) score exceeding 2, a total of 440 differences were discerned from the phylum to OTU level among ErHL-0 (the cecal samples of Er-HL fed with basal diet), LLW-0 (the cecal samples of L-LW fed with basal diet) and SLW-0 (the cecal samples of S-LW fed with basal diet) groups (Figure 5A, Appendix A). At the genera level, Figure 5D enumerates 70 different genera, 22 of which had a higher abundance in the ErHL-0 group. A total of 335 differences (*p* < 0.05, LDA > 2) were discerned from phylum to OTU level among ErHL-7 (the cecal samples of Er-HL fed with 7% WRB), LLW-7 (the cecal samples of L-LW fed with 7% WRB) and SLW-7 (the cecal samples of S-LW fed with 7% WRB) groups (Figure 5B, Appendix A), where 69 different genera were represented in Figure 3E. Among these, 32 genera had a higher abundance in the ErHL-7 group. A total of 594 differences (*p* < 0.05, LDA > 2) were discerned from the phylum to OTU level among ErHL-21 (the cecal samples of Er-HL fed with 21% WRB), LLW-21 (the cecal samples of L-LW fed with 21% WRB) and SLW-21 (the cecal samples of S-LW fed with 21% WRB) groups (Figure 5C, Appendix A). Eighty-five different genera were identified in Figure 3F, with 41 genera being elevated in ErHL-21 group. 

Of particular note, it was observed that in swine subjected to a basal diet, the abundance of *Ruminococcaceae* (Figure 5A, z) was higher in the L-LW group. However, upon dietary modification to include 7% WRB, this family had a higher abundance in the Er-HL group (Figure 5B, x).

### 3.5. Selecting the Key Bacteria Related to the High Fiber Digestibility of Erhualian Pigs

The Er-HL exhibited a higher ATTD of fiber compared with L-LW and S-LW. When comparing the different bacteria among the three breeds, it was found that a total of 22 genera exhibited a higher abundance in the ErHL-0 group, 32 genera exhibited higher abundance in the ErHL-7 group, and 41 genera exhibited higher abundance in the ErHL-21 group relative to the other breeds. Moreover, 24 genera demonstrated heightened abundance in a minimum of two Er-HL groups. Consequently, these 65 distinct genera with heightened abundance in the Er-HL group among the three breeds were postulated as potential key bacteria related to high fiber digestibility of Erhualian pigs (Appendix A). 

The fiber digestibility (cellulose and hemicellulose) in Er-HL fed the basal and 7% WRB diets was significantly higher than that of Er-HL fed the 21% WRB diet. Therefore, taxonomic distinctions at the genera level were scrutinized between Er-HL fed Basal or 7% WRB and those receiving 21% WRB. In the cecal microbiota of Er-HL, 14 different genera were identified between the Basal and 21% WRB groups, with 11 genera exhibiting a higher abundance in the ErHL-0 group (Appendix A). Likewise, 13 genera were different between the 7% WRB and 21% WRB groups, of which six genera exhibited a higher abundance in ErHL-7 group (Appendix A). Thus, 17 genera exhibiting a higher abundance in the Basal and 7% WRB groups were postulated as potential key bacteria related to the high fiber digestibility of Erhualian pigs (Appendix A).

A correlative analysis was conducted encompassing the identified potential key bacteria (65 different genera with higher abundance in Er-HLacross the three breeds and 17 genera with higher abundance in Basal and 7% WRB groups). A positive correlation between the abundance of 44 genera and ATTD of fiber was discerned in the entire porcine sample (Appendix A). Further refinement of this analysis specific to the Er-HL substantiated that 13 genera were positively correlated with ATTD of fiber in Er-HL (Figure 6). These 13 genera were selected as key bacteria related to the high fiber digestibility of Erhualian pigs.

Functional annotations for these 13 key genera were pursued via PICRUSt2 [42]. Given that dietary fiber constitutes a subset of carbohydrates, Kyoto Encyclopedia of Genes and Genomes (KEGG) pathways pertinent to carbohydrate metabolism were scrutinized. These included Glycolysis/Gluconeogenesis, Tricarboxylic Acid (TCA) cycles, Starch and sucrose metabolism, and Pyruvate metabolism (Figure 7). Notably, an enrichment of most carbohydrate metabolism-related pathways was observed in Er-HL samples when compared with S-LW and L-LW samples. 

### 3.6. Potential Core Members of Cecal Microbiome Related to Fiber Digestion and Metabolism in Pigs

SCFA concentrations in cecal content were determined to assess dietary fiber fermentation potential of intestinal microbiota (Table 5). Intriguingly, no discernible variations in SCFA concentrations were observed across Er-HL, S-LW and L-LW irrespective of diets. Elevating the fiber level from Basal to 21% WRB resulted in a concomitant increase in butyrate and valerate concentrations in cecal content, irrespective of breeds (*p* < 0.05).

To refine the identification of microbial entities integral to fiber digestion, a correlational analysis was executed between fiber digestibility and SCFA concentrations, focusing on the primary microbiota delineated by the top 200 OTUs. The analyses revealed that 15 OTUs correlated with acetate concentrations, 5 OTUs with propionate, 7 OTUs with butyrate, and 12 OTUs with total SCFAs (absolute value of spearman correlations > 0.30, *p* < 0.05) (Appendix A). Additionally, significant correlations were observed between 48 OTUs and ATTD of NDF, 48 OTUs with ADF, 42 OTUs with cellulose, 32 OTUs with hemicellulose, 49 OTUs with IDF, and 45 OTUs with TDF (absolute value of spearman correlations > 0.40, *p* < 0.05) (Appendix A). Notably, OTUs 1098, 1199, 1287, 153, 238, 324, 343, 511, 550, 616, 623, 625, 631, 824, and 1006 demonstrated simultaneous and statistically significant correlations with both fiber digestibility and SCFA concentrations in the subject pigs. Upon annotation of these 15 OTUs against the Silva (SSU123) 16S rRNA database, it was ascertained that these OTUs are taxonomically aligned with the genera *Lactobacillus*, *Butyricicoccus*, *Treponema_2*, among 12 other genera.

## 4. Discussion

Chinese local pig breeds, notably the Erhualian pig, are reputed for their tolerance to high-fiber diets [25]. In stark contrast, the Large White pig, a paradigmatic lean-type breed, demonstrates diminished adaptability to high-fiber diets, a vulnerability engendered by extended selective breeding [43]. However, the empirical landscape reveals a paradox: some investigations have reported analogous fiber digestibility between local and lean-type pig breeds [22,23]. This incongruity may be attributable to divergent developmental stages of the porcine subjects during inter-breed comparisons. Prevailing methodologies favor employing pigs of comparable age [44,45] or body weight [20,46] for these analyses. However, given the inherent physiological disparities between local and commercial pig breeds, a myriad of confounding variables may impact the empirical outcomes.

To mitigate such methodological intricacies, our study strategically selected Large White pigs at two distinct developmental stages, synchronizing them in terms of body weight and physiological stage with the Er-HL. These control subjects and Erhualian pigs were subsequently allocated to four fiber-level treatments. This design enabled a nuanced evaluation of fiber tolerance in Chinese local breeds and facilitated the identification of fiber-degrading bacteria related to their high fiber digestibility. To ascertain the adaptability of indigenous breeds to a high-fiber, low-energy diet, wheat bran—a prevalent fiber resource in China—replaced the basal diet directly. Importantly, caloric levels remained unadjusted across treatments, while dietary protein levels were homogenized.

Our findings indicate that fiber level exerted no discernable impact on growth performance (ADFI, ADG) in Er-HL. Conversely, the ADFI and ADG in L-LW decreased when fed with the 7%, 14%, and 21% WRB. This lends empirical support to the high-fiber tolerance of Er-HL as opposed to L-LW. Past studies have reported that Large White pigs exhibit higher ADG in comparison to Chinese local pigs, such as Meishan pigs, of comparable age [47,48]. Concurrently, there were no differences between the ADG in L-LW and Er-HL when fed with the 7%, 14%, and 21% WRB, which means the difference based on genetic background between L-LW and Er-HL was reduced with the increase in dietary fiber. Although no ADG variance was observed between S-LW and Er-HL, that could be due to the age of the pigs and not the breed itself. Additionally, fiber level exerted no discernable impact on ADFI and ADG in S-LW, potentially attributable to the high-fiber content of the basal diet. Neither the National Research Council (NRC) standards nor the Chinese “Pig Raising Standard” specifies the fiber content in pig diet. Consequently, the basal diet was formulated in accordance with the daily dietary fiber content of Erhualian pigs. Thus, this basal diet, comprising 12.84% NDF, which can be used as a high-fiber level in a breed comparison experiment [48], may surpass the tolerance thresholds of S-LW.

In the present study, the ATTD of NDF, ADF, hemicellulose, IDF and TDF were significantly higher in Er-HL relative to L-LW and S-LW, irrespective of diets. This observation underscores the fiber digestibility difference of genetic background between Large White and Erhualian pig. The ATTD of NDF, ADF, IDF and TDF in Er-HL were not influenced by fiber levels, whereas in L-LW, these metrics significantly declined in response to elevated fiber levels. These findings corroborate that Erhualian pigs had a higher fiber digestibility than Large White pigs, by extension, greater tolerance to high-fiber diets. Concurrently, a plethora of studies have reported analogous findings in ancient local pig breeds, including Vietnamese Mong Cai pigs [49], Zimbabwean indigenous Mukota pigs [43], Iberian pigs [19], and Chinese Meishan pigs [50], further substantiating our results. The absence of a breed × diet interaction effect on the ATTD of fiber obviated the need for comparative analysis among breeds to identical dietary regimes. Even though the ATTD of fiber of S-LW was not influenced by various levels of dietary fiber, the ATTD of fiber of S-LW maintained a lower level compared with Er-HL irrespective of diets, suggesting that S-LW have lower tolerance to high-fiber diets. 

Moreover, no discernable difference was evident in the ATTD of cellulose and hemicellulose in Er-HL when fed with 7% diet and basal diet. However, a statistically significant decrement in the ATTD of cellulose and hemicellulose was noted in Er-HL fed with 14% WRB and 21% WRB compared with those of Er-HL fed with 7% WRB. These observations suggest that the fiber tolerance threshold for Erhualian pigs is met at the 7% WRB diet. Beyond this concentration, notably at 14% and 21% WRB, the ATTD of fiber in Erhualian pigs declined, potentially due to the surpassing of tolerance thresholds. This may have perturbed the intestinal milieu, thus mitigating the inherent advantage of Erhualian pigs in fiber digestibility and tolerance.

It is well-established that dietary fiber undergoes microbial fermentation in the large intestine of pigs. In the present study, the ATTD of cellulose and hemicellulose in Er-HL fed with 21% WRB diets decreased significantly compared with those in Er-HL fed with 7% WRB diet. Consequently, we elected to conduct 16S rRNA gene sequencing of cecal contents in Er-HL, L-LW and S-LW pigs fed with the basal, 7% WRB and 21% WRB diet. According to diversity indices, the sequence coverage rate exceeded 99%, bolstering the validity of the findings. Our data revealed that the cecal microbiota of Er-HL exhibited greater richness estimators than L-LW and S-LW, irrespective of diets. This disparity highlights the inherent genetic variances in cecal microbiota structure between Large White and Er-HL breeds, and may partly elucidate the superior fiber digestibility exhibited by Er-HL. Furthermore, the cecal microbiota of Er-HL and L-LW exhibited elevated species diversity compared with S-LW, an observation corroborated by the extant literature indicating that microbiota diversity escalates with the chronological age of swine [18,48]. This discrepancy could be attributable to the divergence in physiological stages between S-LW and the Er-HL and L-LW. Echoing Chen’s study, fiber constituents were found to engender disparate intestinal microbiota profiles [51]. Additionally, escalation in the bran fiber level from Basal to 21% WRB led to a reduction in richness estimators of cecal microbiota. Based on the PCoA and PERMANOVA, we discerned that breed serves as a pivotal determinant in shaping swine gut microbiota, underscoring the influence of host genetics. Irrespective of breed, *Firmicutes* and *Bacteroidetes* remained the predominant phyla, constituting over 85% of the total sequence numbers, a finding in alignment with preceding research on pig intestinal microbiota [52,53].

The principal objective of the sequencing procedure conducted in this investigation was to elucidate specific characteristics of the intestinal microbiota related to the high fiber digestibility of Erhualian pigs. Heinritz et al.’s study substantiates that a diet rich in fiber results in a heightened abundance of *Lactobacilli*, *Bifidobacteria* and *Faecalibacterium prausnitzii* as determined in [54]. Intestinal microbiota and their metabolites are important factors affecting gastrointestinal tract function and health in pigs. The study of Heinritz et al. [54] showed that a high fiber diet stimulated the growth of beneficial bacteria in the intestinal tract. Another study found that proteins unique to the distal swine gut share high sequence homology with known carbohydrate membrane transporters [53]. Among the results of this study, carbohydrate metabolism (13%) was the most abundant in the subsystem. Because of the higher fiber ATTD of Er-HL pigs, the different genera had a higher abundance in the Er-HL group among the three breeds considered as potential key bacteria which were related to the high fiber digestibility of Erhualian pigs. Among them, Anaeroplasma, Norank_f__Clostridiales_vadinBB60_group, Prevotella_1, Ruminiclostridium_6, Ruminococcaceae_UCG-010, Unclassified_f__Erysipelotrichaceae, Unclassified_o_Bacteroidales, and Candidatus_Stoquefichus were all different among the Er-HL, L-LW and S-LW groups when fed with Basal, 7% WRB and 21% WRB diets. These bacteria are more likely to be related to the high fiber digestibility of Erhualian pigs.

Furthermore, we pinpointed 17 genera that could potentially be associated with the metabolism of cellulose and hemicellulose, contingent upon the fiber digestibility of varied dietary fiber levels in Erhualian pigs. A majority of these genera belong to families *Lachnospiraceae* (*Lachnospiraceae_NK4B4_group*, *Acetitomaculum,* etc.) and *Ruminococcaceae* (*Ruminococcaceae_UCG-008, norank_f__Ruminococcaceae*), renowned for their fiber-degrading capabilities [55]. 

In the context of phenotype–microbiota interactions, a total of 44 genera manifested positive correlations with the ATTD of fiber. These include previously described species with fiber-degrading capabilities such as *Lactobacillus* [56], *Fibrobacter* [57], *Cellulosilyticum* [58], and *Ruminiclostridium_6* [59], genera correlated with a high-fiber diet like *Prevotellaceae_UCG-003* [60], *Quinella* [61], *norank_f__Lachnospiraceae* [62], *unclassified_o__Bacteroidales* [63], *Rikenellaceae_RC9_gut_group* [12], *Akkermansia* [64], and genera known for SCFA production like *norank_f_Bacteroidales_S24-7_group* [65].

To identify pivotal bacterial genera that could serve as potential biomarkers related to the high fiber digestibility of Erhualian pigs, an in-depth correlation analysis was conducted between the abundance of 44 genera and ATTD of fiber in Er-HL. Among these, 13 genera demonstrated a positive correlation with ATTD of fiber in Er-HLand and were subsequently designated as key bacteria that could potentially serve as biomarkers related to the high fiber digestibility of Erhualian pigs. Within this selection of key bacteria, *Bacteroides* stood out as a predominant genus in the porcine gut microbiota, as corroborated by the present investigation. Certain species within the *Bacteroides* spp. have been characterized as cellulolytic organisms, such as *Bacteroides succinogenes* [66]. Moreover, swine with an enterotype markedly enriched in Bacteroides were reported to possess a higher copy number of terminal genes responsible for butyrate synthesis, including butyrate synthesis, butyrate kinase and butyryl coenzyme A: acetate CoA transferase than pigs with other enterotype [67]. Several of the 13 identified key bacteria have been previously reported to be associated with high-fiber diets, such as *Prevotella_1* [68,69], *Erysipelotrichaceae_UCG-004* [62], *Hydrogenoanaero bacterium* [70], *dgA-11_gut_group* [71], *hoa5-07d05_gut_group* [72], *Lachnospiraceae_XPB1014_group* [73], *Papillibacter* [61], *Prevotellaceae_UCG-004* [71]. Additionally, genera like *Dielma, norank_f__Erysipelotrichaceae*, *norank_f__Clostridiales_vadinBB60_group* and *unclassified_o__Clostridiales* were newly identified as being associated with high fiber digestibility.

To assess the potential for dietary fiber fermentation by intestinal microbiota, concentrations of SCFAs within the cecal content were quantified. Dietary fiber is known to modulate SCFA concentrations within the porcine gut. Consistent with this notion, Heinritz et al. reported increased production of SCFAs, particularly butyric acid, in pigs subjected to a high-fiber diet [54]. Similarly, Pu et al. found that the concentrations of acetate, propionate, isobutyrate and total SCFA in porcine cecum significantly increased with the dietary fiber level [62]. These observations are congruent with our findings. Although SCFA concentrations have been reported to be lower in local pig breeds compared with commercial breeds, such as Large White pigs [44] and Landrace pigs [74], our study revealed no significant differences in SCFA concentrations in the cecal content among Er-HL, S-LW and L-LW irrespective of diets. This suggests that the high fiber digestibility of Er-HL may offset any differences in SCFAs concentrations in cecal content between Er-HL and Large White pigs, potentially mediated by the differential SCFA-absorptive capacity of local breed colonocytes compared with Large White pig colonocytes [74].

## 5. Conclusions

In summary, Er-HL exhibited high-fiber tolerance both in terms of growth performance and fiber digestibility compared with Large White pigs. Specifically, the ATTD of NDF, ADF, hemicellulose, IDF and TDF were significantly higher in Er-HL compared with L-LW and S-LW, irrespective of diets. Fiber level exerted no discernable impact on growth performance (ADFI, ADG) and the ATTD of fiber (NDF, ADF, IDF and TDF) in Er-HL. Notably, a statistically significant decrement in the ATTD of cellulose and hemicellulose was noted in Er-HL fed with 14% WRB and 21% WRB compared with those of Er-HL fed with 7% WRB. Hence, it is inferred that the optimum fiber level of the Er-HL was identified as 7% WRB (TDF 16.32%). Breed serves as a pivotal determinant in shaping swine gut microbiota. Thirteen genera were ascertained to significantly contribute to high fiber digestibility of Er-HL, correlating with an enhancement of carbohydrate metabolism pathways.

## Figures and Tables

**Figure 1 microorganisms-11-02474-f001:**
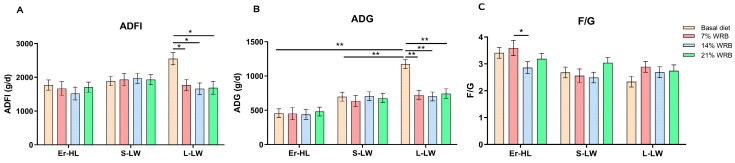
Effects of breeds and dietary fiber levels on ADFI (**A**), ADG (**B**) and F/G (**C**) of pigs. * Means significant difference between two groups (*p* < 0.05). ** Means extreme significant difference between two groups (*p* < 0.01). Values are mean ± standard error. ADFI = average daily feed intake; ADG = average daily gain; F/G = feed: gain ratio. 7% WRB = 7%-wheat-bran-replaced basal diet; 14% WRB = 14%-wheat-bran-replaced basal diet; 21% WRB = 21%-wheat-bran-replaced basal diet.

**Figure 2 microorganisms-11-02474-f002:**
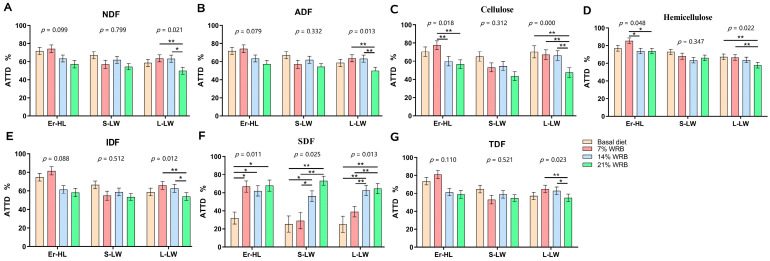
Effects of breeds and dietary fiber levels on apparent total tract digestibility (ATTD) of NDF (**A**), ADF (**B**), cellulose (**C**), hemicellulose (**D**), IDF (**E**), SDF (**F**), TDF (**G**) in pigs. * Means significant difference between two groups (*p* < 0.05). ** Means significant difference between two groups (*p* < 0.01). Values are mean ± standard error. NDF = neutral detergent fiber; ADF = acid detergent fiber; IDF = insoluble dietary fiber; SDF = soluble dietary fiber; TDF = total dietary fiber. 7% WRB = 7%-wheat-bran-replaced basal diet; 14% WRB = 14%-wheat-bran-replaced basal diet; 21% WRB = 21%-wheat-bran-replaced basal diet.

**Figure 3 microorganisms-11-02474-f003:**
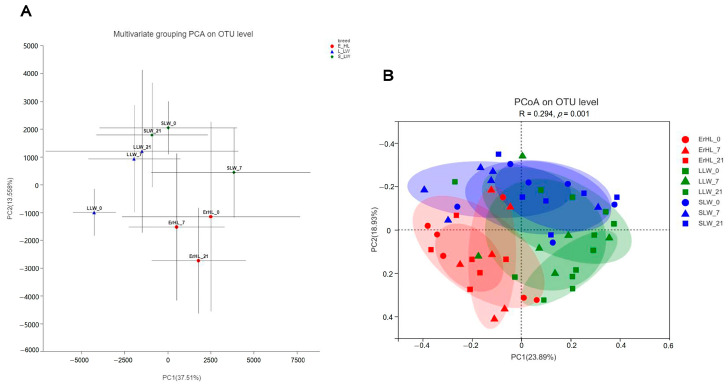
Effects of breeds and dietary fiber levels on Beta-diversity of cecal microbiota of pigs. (**A**) Er_HL = the cecal samples of Erhualian fattening barrows; L_LW = the cecal samples of large Large White fattening barrows; S_LW = the cecal samples of small Large White fattening barrows. (**B**) ErHL_0 = the cecal samples of Erhualian fattening barrows fed with basal diet; ErHL_7 = the cecal samples of Erhualian fattening barrows fed with 7%-wheat-bran-replaced basal diet; ErHL_21 = the cecal samples of Erhualian fattening barrows fed with 21%-wheat-bran-replaced basal diet; LLW_0 = the cecal samples of large Large White pigs fed with basal diet; LLW_7 = the cecal samples of large Large White pigs fed with 7%-wheat-bran-replaced basal diet; LLW_21 = the cecal samples of large Large White pigs fed with 21%-wheat-bran-replaced basal diet; SLW_0 = the cecal samples of small Large White fattening barrows fed with basal diet; SLW_7 = the cecal samples of small Large White fattening barrows fed with 7%-wheat-bran-replaced basal diet; SLW_21 = the cecal samples of small Large White fattening barrows fed with 21%-wheat-bran-replaced basal diet.

**Figure 4 microorganisms-11-02474-f004:**
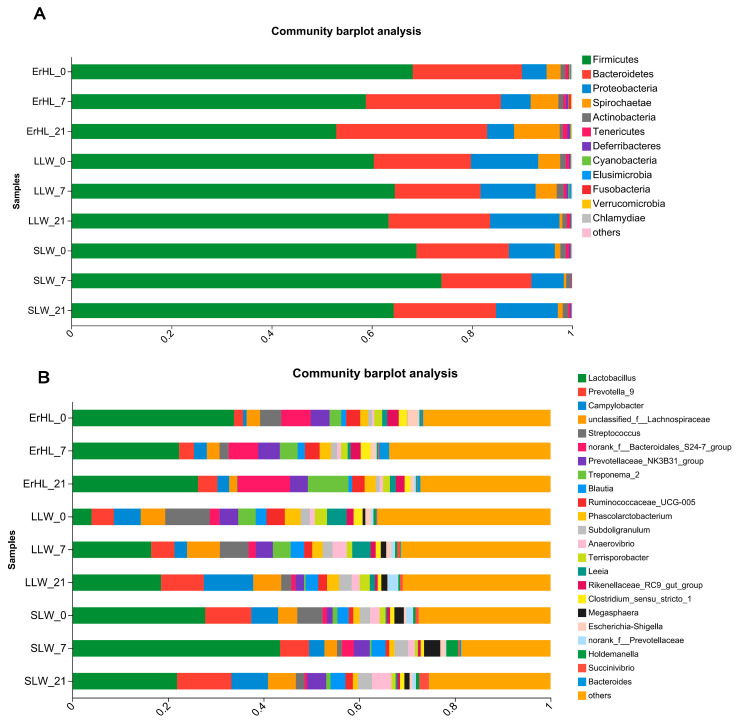
Phyla (**A**) and genera (**B**) distribution of cecal microbiota of pigs. ErHL_0 = the cecal samples of Erhualian fattening barrows fed with basal diet; ErHL_7 = the cecal samples of Erhualian fattening barrows fed with 7%-wheat-bran-replaced basal diet; ErHL_21 = the cecal samples of Erhualian fattening barrows fed with 21%-wheat-bran-replaced basal diet; LLW_0 = the cecal samples of large Large White pigs fed with basal diet; LLW_7 = the cecal samples of large Large White pigs fed with 7%-wheat-bran-replaced basal diet; LLW_21 = the cecal samples of large Large White pigs fed with 21%-wheat-bran-replaced basal diet; SLW_0 = the cecal samples of small Large White fattening barrows fed with basal diet; SLW_7 = the cecal samples of small Large White fattening barrows fed with 7%-wheat-bran-replaced basal diet; SLW_21 = the cecal samples of small Large White fattening barrows fed with 21%-wheat-bran-replaced basal diet.

**Figure 5 microorganisms-11-02474-f005:**
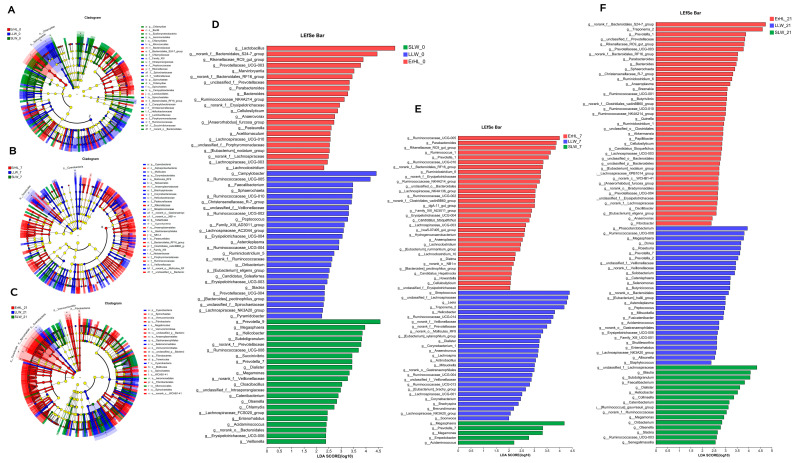
Linear discriminant analysis effect size (LEfSe) on genera level of cecal microbiota. (**A**) Differences from phylum to OTU level among ErHL-0, LLW-0 and SLW-0 groups. ErHL_0 = the cecal samples of Erhualian fattening barrows fed with basal diet; LLW_0 = the cecal samples of large Large White pigs fed with basal diet; SLW_0 = the cecal samples of small Large White fattening barrows fed with basal diet. (**B**) Differences from phylum to OTU level among ErHL-7, LLW-7 and SLW-7 groups. ErHL_7 = the cecal samples of Erhualian fattening barrows fed with 7%-wheat-bran-replaced basal diet; LLW_7 = the cecal samples of large Large White pigs fed with 7%-wheat-bran-replaced basal diet; SLW_7 = the cecal samples of small Large White fattening barrows fed with 7%-wheat-bran-replaced basal diet. (**C**) Differences from phylum to OTU level among ErHL-21, LLW-21 and SLW-21 groups. ErHL_21 = the cecal samples of Erhualian fattening barrows fed with 21%-wheat-bran-replaced basal diet; LLW_21 = the cecal samples of large Large White pigs fed with 21%-wheat-bran-replaced basal diet; SLW_21 = the cecal samples of small Large White fattening barrows fed with 21%-wheat-bran-replaced basal diet. (**D**) Different genera among ErHL-0, LLW-0 and SLW-0 groups. (**E**) Different genera among ErHL-7, LLW-7 and SLW-7 groups. (**F**) Different genera among ErHL-21, LLW-21 and SLW-21 groups.

**Figure 6 microorganisms-11-02474-f006:**
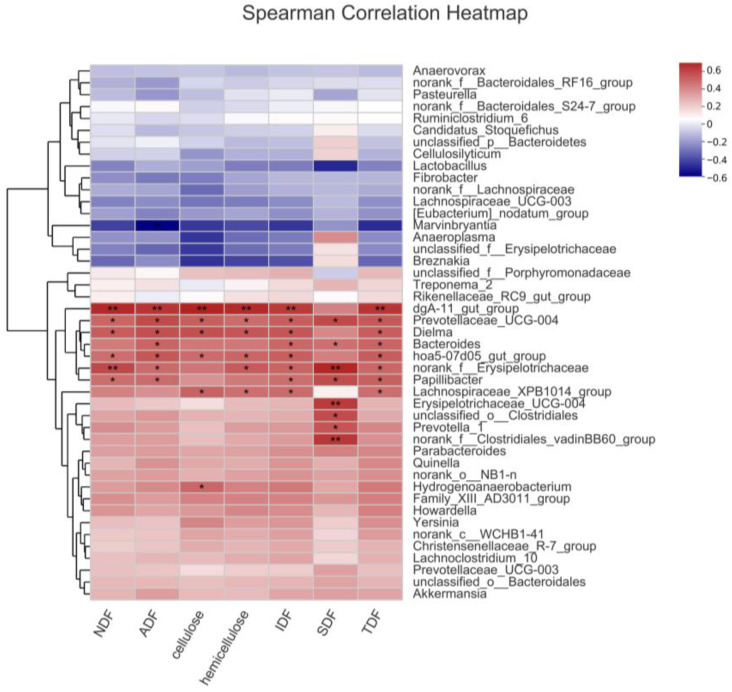
Correlation heatmap between potential key bacteria abundance and fiber apparent total tract digestibility of Erhualian pigs. * Means significant correlation (*p* < 0.05). ** Means significant correlation (*p* < 0.01).

**Figure 7 microorganisms-11-02474-f007:**
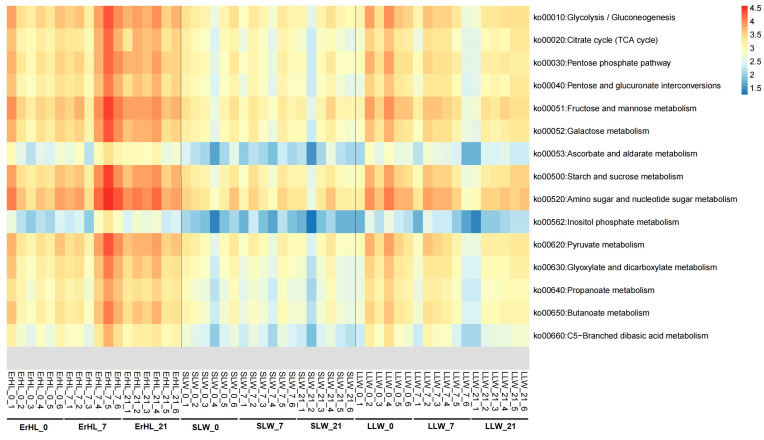
Distribution of carbohydrate-metabolism-related pathways of key bacteria. ErHL_0 = the cecal samples of Erhualian fattening barrows fed with basal diet; ErHL_7 = the cecal samples of Erhualian fattening barrows fed with 7%-wheat-bran-replaced basal diet; ErHL_21 = the cecal samples of Erhualian fattening barrows fed with 21%-wheat-bran-replaced basal diet; LLW_0 = the cecal samples of large Large White pigs fed with basal diet; LLW_7 = the cecal samples of large Large White pigs fed with 7%-wheat-bran-replaced basal diet; LLW_21 = the cecal samples of large Large White pigs fed with 21%-wheat-bran-replaced basal diet; SLW_0 = the cecal samples of small Large White fattening barrows fed with basal diet; SLW_7 = the cecal samples of small Large White fattening barrows fed with 7%-wheat-bran-replaced basal diet; SLW_21 = the cecal samples of small Large White fattening barrows fed with 21%-wheat-bran-replaced basal diet.

**Table 1 microorganisms-11-02474-t001:** Effects of breeds and dietary fiber levels on body weight, average daily feed intake (ADFI), average daily gain (ADG), feed/gain ratio (F/G) of pigs.

Item ^1^	Breeds ^2^			Diets ^3^				RMSE	*p*-Values		
	Er-HL	L-LW	S-LW	Basal	7% WRB	14% WRB	21% WRB		Breeds	Diets	Breeds × Diets
Initial body weight (kg)	44.76 ^B^	68.29 ^A^	46.76 ^B^	/	/	/	/	1.38	0.000	/	/
Final body weight (kg)	57.57 ^C^	91.70 ^A^	65.80 ^B^	75.60 ^A^	71.73 ^AB^	67.37 ^B^	72.05 ^AB^	3.70	0.000	0.053	0.146
ADFI (kg/d)	1.67	1.92	1.93	2.07 ^a^	1.79 ^ab^	1.71 ^b^	1.78 ^ab^	0.16	0.056	0.015	0.012
ADG (g/d)	457.56 ^C^	835.81 ^A^	680.01 ^B^	776.52 ^a^	603.97 ^b^	615.37 ^b^	635.32 ^ab^	70.42	0.000	0.008	0.003
F/G	3.27 ^A^	2.66 ^B^	2.69 ^B^	2.81	3.01	2.68	2.99	0.22	0.001	0.236	0.195

^1^ Values are least square mean; RMSE means Root Mean Square Error. ^abc^ Means rows with different letters are significantly different (*p* < 0.05). ^ABC^ Means rows with different letters are significantly different (*p* < 0.01). ^2^ Er-HL = Erhualian fattening barrow; L-LW = Large White fattening barrows, the same physiological stage as Erhualian pigs; S-LW = small Large White fattening barrows, the same body weight as Erhualian pigs. ^3^ Basal = Basal diet; 7% WRB = 7%-wheat-bran-replaced basal diet; 14% WRB = 14%-wheat-bran-replaced basal diet; 21% WRB = 21%-wheat-bran-replaced basal diet. ^a–c^ Within a row, values with different superscript letters differ (*p* < 0.05).

**Table 2 microorganisms-11-02474-t002:** Effects of breeds and dietary fiber levels on apparent total tract digestibility in pigs.

Item ^1^	Breeds ^2^			Diets ^3^				RMSE	*p*-Values		
	Er-HL	L-LW	S-LW	Basal	7% WRB	14% WRB	21% WRB		Breeds	Diets	Breeds × Diets
NDF	66.08 ^A^	52.53 ^B^	55.77 ^B^	60.40 ^AB^	63.31 ^A^	57.85 ^AB^	50.95 ^B^	6.90	0.001	0.013	0.428
ADF	66.40 ^Aa^	56.78 ^Bb^	59.73 ^ABb^	63.78 ^ABa^	64.12 ^Aa^	62.08 ^ABab^	53.90 ^Bb^	5.74	0.004	0.003	0.320
Cellulose	66.20 ^A^	62.83 ^AB^	54.13 ^B^	68.63 ^Aa^	66.00 ^Aa^	60.28 ^ABa^	49.29 ^Bb^	6.73	0.000	0.000	0.212
Hemicellulose	77.60 ^A^	63.90 ^B^	67.64 ^B^	72.39 ^A^	73.38 ^A^	67.11 ^B^	65.98 ^B^	4.83	0.000	0.010	0.365
IDF	68.55 ^A^	57.08 ^B^	57.61 ^B^	63.42 ^ab^	65.82 ^a^	59.70 ^ab^	55.38 ^b^	6.01	0.000	0.015	0.104
SDF	58.05	44.48	47.23	27.55 ^C^	44.09 ^BC^	58.81 ^AB^	69.23 ^A^	8.47	0.033	0.000	0.085
TDF	71.61 ^A^	56.00 ^B^	56.24 ^B^	61.16	65.75	59.87	58.35	5.98	0.000	0.169	0.06
EE	84.51	80.75	81.05	84.41 ^a^	83.61 ^ab^	81.33 ^ab^	79.06 ^b^	3.16	0.043	0.018	0.074
CP	85.64 ^Aa^	84.64 ^ABa^	80.66 ^Bb^	87.94 ^Aa^	82.84 ^ABb^	82.67 ^Bab^	81.14 ^Bab^	2.78	0.001	0.001	0.036

^1^ Values are least square mean; RMSE means Root Mean Square Error. ^abc^ Means rows with different letters are significantly different (*p* < 0.05). ^ABC^ Means rows with different letters are significantly different (*p* < 0.01). NDF = neutral detergent fiber; ADF = acid detergent fiber; IDF = insoluble dietary fiber; SDF = soluble dietary fiber; TDF = total dietary fiber; EE = ether extract; CP = crude protein. ^2^ Er-HL = Erhualian fattening barrow; L-LW = Large White fattening barrows, the same physiological stage as Erhualian pigs; S-LW = small Large White fattening barrows, the same body weight as Erhualian pigs. ^3^ Basal = Basal diet; 7% WRB = 7%-wheat-bran-replaced basal diet; 14% WRB = 14%-wheat-bran-replaced basal diet; 21% WRB = 21%-wheat-bran-replaced basal diet. ^a^^–c^ Within a row, values with different superscript letters differ (*p* < 0.05).

**Table 3 microorganisms-11-02474-t003:** Effects of breeds and dietary fiber levels on diversity and abundance of cecal microbiota in pigs.

Item ^1^	Breeds ^2^			Diets ^3^			RMSE	*p*-Values		
	Er-HL	L-LW	S-LW	Basal	7% WRB	21% WRB		Breeds	Diets	Breeds × Diets
Chao	886.93 ^Aa^	847.24 ^ABb^	725.22 ^Cc^	889.03 ^a^	813.33 ^b^	757.02 ^c^	44.33	0.002	0.009	0.448
Ace	890.4 ^Aa^	829.79 ^ABb^	734.58 ^Cc^	882.49 ^a^	815.47 ^b^	756.81 ^c^	42.12	0.002	0.020	0.303
Shannon	4.24 ^a^	4.36 ^a^	3.88 ^b^	4.28	4.10	4.09	0.17	0.010	0.363	0.058
Simpson	21.85	11.13	11.13	13.98	12.42	10.6	1.10	0.057	0.896	0.001

^1^ Values are least square mean; RMSE means Root Mean Square Error. ^abc^ Means rows with different letters are significantly different (*p* < 0.05). ^ABC^ Means rows with different letters are significantly different (*p* < 0.01). ^2^ Er-HL = Erhualian fattening barrow; L-LW = Large White fattening barrows, the same physiological stage as Erhualian pigs; S-LW = small Large White fattening barrows, the same body weight as Erhualian pigs. ^3^ Basal = Basal diet; 7% WRB = 7%-wheat-bran-replaced basal diet; 14% WRB = 14%-wheat-bran-replaced basal diet; 21% WRB = 21%-wheat-bran-replaced basal diet.

**Table 4 microorganisms-11-02474-t004:** PERMANOVA analysis of the factors affecting the gut microbiota of pigs.

Characteristics	Df	SumsOfSqs	MeanSqs	F_Model	R2	Pr (>F)
Breed	2	1.3	0.65	3.92	0.13	0.0001
Diet	2	0.33	0.17	1	0.03	0.4254
Body weight	1	0.1	0.1	0.6	0.01	0.8485
Average daily feed intake	1	0.2	0.2	1.2	0.02	0.2648
Residual	46	7.65	0.17		0.74	
Total	52	10.35			1	

**Table 5 microorganisms-11-02474-t005:** Effects of breeds and dietary fiber levels on SCFA concentrations (µmol/g digesta) in cecal content in pigs.

Item ^1^	Breeds ^2^			Diets ^3^				RMSE	*p*-Values		
	Er-HL	L-LW	S-LW	Basal	7% WRB	14% WRB	21% WRB		Breeds	Diets	Breeds × Diets
Acetate	241.67	300.60	185.08	235.17	215.45	278.37	240.81	33.84	0.08	0.54	0.16
Propionate	99.39	101.93	81.22	84.76	82.86	120.08	89.03	13.22	0.42	0.10	0.21
Butyrate	40.13	38.61	39.66	32.00 ^Bb^	34.40 ^Bb^	56.10 ^Aa^	35.36 ^ABb^	5.63	0.98	0.00	0.27
Valerate	7.65	7.80	6.51	5.33 ^Bb^	7.45 ^ABb^	10.25 ^Aa^	6.26 ^ABab^	1.21	0.67	0.01	0.17
Total SCFAs	397.80	457.49	317.14	365.61	346.08	473.31	378.24	50.38	0.16	0.21	0.20

^1^ Values are least square mean, RMSE means Root Mean Square Error. ^ab^ Means rows with different letters are significantly different (*p* < 0.05). ^AB^ Means rows with different letters are significantly different (*p* < 0.01). ^2^ Er-HL = Erhualian fattening barrow; L-LW = Large White fattening barrows, the same physiological stage as Erhualian pigs; S-LW = small Large White fattening barrows, the same body weight as Erhualian pigs. ^3^ Basal = Basal diet; 7% WRB = 7%-wheat-bran-replaced basal diet; 14% WRB = 14%-wheat-bran-replaced basal diet; 21% WRB = 21%-wheat-bran-replaced basal diet.

## Data Availability

The datasets of 16S rRNA gene sequence for this study can be found in the NCBI PRJNA889677: https://www.ncbi.nlm.nih.gov/sra/PRJNA889677 (accessed on 13 October 2022).

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
