# Peer review of "Effects of Varying Levels of Wheat Bran Dietary Fiber on Growth Performance, Fiber Digestibility and Gut Microbiota in Erhualian and Large White Pigs"

_microorganisms, 2023, doi:10.3390/microorganisms11102474_

Round 1
Reviewer 1 Report
In this manuscript, the authors compared the effect of varying levels of dietary fibers in the diet on growth performance, digestibility and microbiota in a local Chinese breed and a high productive breed. Their results suggest that Erhualian pigs have higher fiber digestibility and maintain growth performance with high fiber diets compared to Large White pigs.
This is manuscript address an important issue for pig production, which is the availability of feed resources and the potential of local breeds in this context. However, there are several concerns that should be taken into consideration before this is considered for publication. I have two general concerns (the first two comments) and I will then list more detailed and minor issues.
General comments:
1. This paper would benefit from some closer proof reading, both regarding typos (for example, the first sentence of the introduction has a typo “The At present…”) and quality of English language. A proof reading by a professional English language editor or native English speaker would greatly improve the manuscript.
2. The authors chose to compare a local Chinese breed to 2 groups of LW : one with same body weight and one with same physiological stage as Erhualian pigs. This is a very relevant experimental design as there are important differences in maturity and growth in the 2 breeds and it is important to investigate whether the observed effect in Erhualian pigs could be due to body weight or physiological stage rather than genetic background. However, the authors never discuss this point as such, despite the fact that some results are very similar between Erhualian pigs and S-LW pigs (growth performance and fiber digestibility is maintained at high fiber levels in both groups). This is also the case for cecal microbiota diversity, which is similar between Erhualian and L-LW, suggesting an influence of physiological stage. It seems important to highligh this point in the discussion.
Specific comments:
3. In the abstract, some results have P-values while other don’t. Please add the missing P-values.
4. L30-33: The sentence is not clear and should be reformulated.
5. In the introduction, the authors mention the importance of increasing our use of non-conventional resources for economic reasons, it would be good to also mention the important carbon footprint of corn and soybean meal.
6. Also in the introduction, it would be good to add some background on Erhualian pigs: what are their characteristics (growth, meat quality, profilacy)? Where do they come from and what is their economic importance in China?
7. L63-70: This is a very detailed description of the different bacteria in the pig intestine. I don’t think that it should be in the introduction but rather in the discussion section.
8. L94-97: The sentence is not clear and should be reformulated. The authors mention “Chinese pig breeds” (plural) but there is only one Chinese breed in the study. “2 types of Large white pig”: the use of “type” can be confusing here. It could mean that the 2 groups of Large White had different genetic characteristic even though only the body weight changes.
9. Throughout the manuscript “Large White” is sometimes written with upper cases at the beginning of each word and sometimes not. Please use the standard writing: Large White or the abbreviation LW.
10. L113: “12 groups were allotted for 3 breeds and 4 treatment, and 6 replicates for each group”: This sentence is not clear should be reformulated, a different word than “breed” should be used as it is not 3 different breeds but 2.
11. L117-120: “The pigs of each group were individually housed in pens” : Were the pigs housed in individual pens as suggested by this sentence or each group of 6 pigs was in a pen (which seems more logical with the use of feeding station)? Also, please specify that Osbourne Testing stations are automatic feeders allowing to record ADFI and weight gain.
12. L120-121: You mention that the temperature and humidity was kept constant, as both parameters can greatly influence growth performance, could you specify at which level was the temperature and humidity set up and the actual measures if they are available.
13. L196-199: Regarding the statistical model, the effect of slaughter batches is included in the model but was not mention before. If the pigs were slaughtered in different batches, it should be mentioned before. Also, I suppose that final body weight is included as a covariate, the phrasing “Wl is the represent effect” is not clear.
14. In the result section, several sentences refer to breed or diet effects, to make it clearer in the text, it would be good to add “irrespective of breeds” for diet effects and “irrespective of diet” for breed effects. For example, L220-222.
15. L276-279: Sentence should be reformulated: it is not clear that the main difference between breeds is the 7% diet.
16. L464-475: This is a repetition of the experimental design that is not necessary in the discussion section. One sentence to summarize the aim of the experiment should be enough.
17. L496-499: “We also found 495 that the ATTD of fiber of S-LW was not influenced by various levels of dietary fiber, either, which might be because of the high fiber content of the basal diet. The basal diet was 497 formulated according to Chinese “Pig Raising Standard”, so S-LW had low ATTD of fiber even feed with basal diet.” Please explain to which ATDD you refer to in these sentences. If it is the ATTD of cellulose and hemicellulose, according to figure 2, the ATTD of S-LW for the basal diet is not lower than in L-LW. Since S-LW also have good performance results with high fiber diet, could it be that the body weight plays a role in the tolerance to high fiber diet (small pigs would be more tolerant that large pigs) ? (See comment n°2)
18. L501: “Our result showed that Er-HL fed with 21% WRB diets decreased significantly comparing with those of Er-HL fed with 7% WRB diet.” What is decreasing? ATTD?
19. “And the cecal microbiota of Er-HL and L-LW pigs had higher species diversity than those of S-LW pigs.” This could suggest that the diversity of the cecal microbiota is influenced by physiological stage (see comment n°2).
This paper would benefit from some closer proof reading, both regarding typos (for example, the first sentence of the introduction has a typo “The At present…”) and quality of English language. A proof reading by a professional English language editor or native English speaker would greatly improve the manuscript.
Reviewer 2 Report
Abstract
- L 32: replace “a enrichment” with “an enrichment”.
Introduction
More recent literature review is needed. Please revised your references since more than 60 % are studies conducted before 2015.
- L 37: “The At present”. Please correct.
- L 39: please add a reference.
Materials and Methods
I suggest to add the ingredient composition of experimental diets (Table S1) in Materials and Methods as Table 1.
- L 114: you added the reference [24]. The experimental design of this study has been published in another article? Or did you used the protocol described in the cited reference?
- L 118: add the dimensions of pens.
- L 120: please mention the producer of the equipment (city, country)
- L 121: please add the values for temperature and humidity.
- L 125-127: “And the composition, estimated nutritional value and determination analysis of the diets were shown in Table S1.” Please rephrase.
- L 149: Hemicellulose and cellulose were calculated using equations based on NDF and ADF? If so, please mention it in this part.
- L 150: please add the equipment manufacturer (city, country)
- L 151: please remove “And” at the beginning of the sentence.
- L 153: replace “apparatus” with equipment
- L 153: add the equipment manufacturer (city, country) used for ether extract determination
- L 157: add the GC model and manufacturer (city, country) used for determination of SCFAs concentrations.
- L 159: add a brief description of the method used for SCFAs analysis - sample preparation and chromatographic conditions (column, mobile phase, flow, temperature, wavelength …).
- L 159: the quantification of the SCFAs has been made based on individual calibration curves?
- L 181: please remove “And” at the beginning of the sentence.
Results
- L 239: please finish the sentence and after add the figure.
- L 414: the reference should be mention with a number.
- L 435: the table number is 5!
- L 432: correct “amang” with “among”
- L 436: add the measurement unit for the SCFAs concentrations
- L 445-446: the sentence is not clear. Please rephrase.
Discussion
- L 471-475: this information is already presented in Materials and Methods
- L 490: mention P value
- L 497-498: this information should be moved to Materials and Methods
- L 524: please discus the SCFAs results
- L 562-564: the cited reference is not relevant to this study, since is presenting results in ruminants.
- L 578: replace “a enrichment” with “an enrichment”.
Round 2
Reviewer 1 Report
I appreciate the authors’ revisions, especially concerning the methods and discussion section that has been improved. However, some parts of the manuscript still need clarification and the manuscript would benefit from a closer proof-reading, especially in the sections added during the revision.
L129: Please remove “individually” since the pigs are housed in groups of 6.
L505-516: The authors argue that the fact that S-LW have similar ADG and ADFI that Er-HL pigs when fed the basal diet is due to the composition of the diet, but if the basal diet is based on Chinese Pig Raising Standard, why would it have a high fiber content? The authors say that “In general, Large White pigs have higher ADG compared to Erhualian pigs” but I suppose it is true when Er-HL and LW are compared at same age? Is it also the case when they are compared at same BW? The reference (46) given here do not give any indication regarding this fact, so it should be changed. My point is that you cannot rule out that the difference in ADG and ADFI could be due to the body weight of the pigs and not the breed itself. If you do not have additional data to prove or refute this point, this hypothesis should be mentioned in the discussion.
L530-533: “So, we did not say that S-LW have the tolerance to high fiber diets”. I understand your arguments here, however the last sentence should be reformulated as “Even though the ATTD of fiber of S-LW was not influenced by various levels of dietary fiber, the ATTD of fiber of S-LW maintained a lower level compared with Er-HL irrespective of diets, suggesting that S-LW have lower tolerance to high fiber diets”.
L531: Why is there a reference here (26), since you are talking about the results of the present study?
L636-639: “Er-HL showed a fiber tolerance in terms of growth performance and fiber digestibility compared with Large White pigs, the ATTD of NDF, ADF, hemicellulose, IDF and TDF were significantly higher in Er-HL than in L-LW and S-LW irrespective of diets, increasing the fiber level did not affect growth performance (ADFI, ADG) and ATTD of fiber (NDF, ADF, IDF and TDF) of Er-HL”. This should be 3 different sentences.
There are many English errors in the manuscript and the language register is sometimes not appropriate for a scientific paper (extensive use of "so", "So, we did not say that..."(L533)...). Therefore, I reiterate my comment that a proof reading by a professional English language editor or native English speaker would greatly improve the manuscript.
